# A qPCR Assay for the Fast Detection and Quantification of *Colletotrichum lupini*

**DOI:** 10.3390/plants10081548

**Published:** 2021-07-28

**Authors:** Tim Kamber, Nachelli Malpica-López, Monika M. Messmer, Thomas Oberhänsli, Christine Arncken, Joris A. Alkemade, Pierre Hohmann

**Affiliations:** 1Department of Crop Sciences, Research Institute of Organic Agriculture (FiBL), Ackerstrasse 113, 5070 Frick, Switzerland; tim.kamber@fibl.org (T.K.); monika.messmer@fibl.org (M.M.M.); thomas.oberhaensli@fibl.org (T.O.); christine.arncken@fibl.org (C.A.); joris.alkemade@fibl.org (J.A.A.); 2Department of Environmental Sciences, University of Basel, Bernoullistrasse 30/32, 4056 Basel, Switzerland; nachelli.malpica@unibas.ch

**Keywords:** anthracnose, *Colletotrichum* *acutatum* species complex, *Colletotrichum* *lupini*, hydrolysis probe, lifecycle, *Lupinus albus*, qPCR, seed

## Abstract

White lupin (*Lupinus albus*) represents an important legume crop in Europe and other parts of the world due to its high protein content and potential for low-input agriculture. However, most cultivars are susceptible to anthracnose caused by *Colletotrichum lupini,* a seed- and air-borne fungal pathogen that causes severe yield losses. The aim of this work was to develop a *C. lupini-*specific quantitative real-time TaqMan PCR assay that allows for quick and reliable detection and quantification of the pathogen in infected seed and plant material. Quantification of *C*. *lupini* DNA in dry seeds allowed us to distinguish infected and certified (non-infected) seed batches with DNA loads corresponding to the disease score index and yield of the mother plants. Additionally, *C. lupini* DNA could be detected in infected lupin shoots and close to the infection site, thereby allowing us to study the disease cycle of this hemibiotrophic pathogen. This qPCR assay provides a useful diagnostic tool to determine anthracnose infection levels of white lupin seeds and will facilitate the use of seed health assessments as a strategy to reduce the primary infection source and spread of this disease.

## 1. Introduction

White lupin (*Lupinus albus* L.) belongs to the Fabaceae family (legumes); it is cultivated worldwide and could play an important role in sustainable agricultural farming systems. Its high seed protein content (average 35%) and the nitrogen-fixing capacity of root-associated bacteria make white lupin an excellent local alternative to soybean and a valuable element of sustainable crop rotations [1,2,3]. With its tolerance to various abiotic stresses, lupin is also considered to be a viable option for the recovery of poor and contaminated soils [4,5].

Anthracnose is a global threat to white lupin cultivation and is considered a main obstacle to its broader agricultural use, as most white lupin cultivars are susceptible to the disease. In the past, the causal agent was thought to be *Colletotrichum gloeosporioides*, however, these isolates were reassigned as a new species designated as *C. lupini* [6,7]. The current anthracnose outbreak is caused by globally distributed, highly aggressive *C. lupini* strains of genetic group II [8,9]. Severe outbreaks of *C*. *lupini* mainly occur in temperate zones under warm and humid weather conditions and can lead to high yield losses [10,11]. Typical symptoms of the disease are closed and wrinkled leaves, twisting and bending stems, and lesions on the stems, leaves, and pods [8,9,11]. Environmental factors, including temperature and rainfall, influence the rate of anthracnose infection and spread under field conditions [12,13]. While secondary infections can spread rapidly by rain splashes, the primary infection route is via seeds [10,14]. The pathogen can cause almost total yield loss when the infection is severe and left untreated [10]. The mechanism by which *Colletotrichum* species are able to survive in the seeds has not yet been completely elucidated, and seed treatments, although encouraging, have proven to be only partially effective in eliminating the pathogen at this stage [15,16,17]. The detection of the pathogen in symptomless but infected seeds could improve seed quality management and, thus, help to reduce disease outbreaks in the field.

A culture-based method to quantify the *C. lupini* infection rate of lupin seeds is the appressoria test, where seeds are placed on poor nutrition media and appressoria are identified microscopically after ten days of incubation [18]. Another method includes a precultivation step to enrich for fungal biomass and to detect the presence or absence of *C*. *lupini* by conventional PCR [19]. For both culture- and PCR-based methods, the severity of infection and detection sensitivity of each seed remain unclear. A promising technique for quantifying latent infections is quantitative real-time PCR (qPCR). Besides saving time and its ease of use, qPCR offers increased detection sensitivity and quantitative information of infected material [20]. Additionally, qPCR can be applied as a high-throughput method for mass screening as many samples can be analyzed in parallel. Thus, the aim of this study was to develop a qPCR assay for the detection and quantification of the pathogen in white lupin seeds and to establish its relation to disease severity and yield of the mother plants.

## 2. Results

### 2.1. Quantification of C. lupini DNA from Pure Cultures

Preliminary tests using internal transcribed spacer regions (ITS) of *C. lupini* as targets for qPCR assays indicated that they are unsuitable as they showed unspecific results when using DNA from non-target species. The glyceraldehyde-3-phosphate dehydrogenase gene (GAPDH) was identified as a suitable target as the sequence showed a high sequence diversity between relevant sister taxa that could co-occur with *C. lupini* on host plants (Appendix A). The TaqMan qPCR assay showed amplification of the GAPDH gene of *C. lupini*, while DNA of the other tested *Colletotrichum* spp. of the *C. acutatum* species complex—*C. tamarilloi* (clade 1), *C*. *nymphaeae* CBS 130239 (clade 2), *C*. *acutatum* (clade 4), and two more distant species *C*. *coccodes* and *C*. *trifolii*—was not amplified (Table 1). The amplification efficiency of 10-fold dilutions of *C. lupini* DNA was 1.06 (y = −3.19 × log(conc) + 28.40; R^2^ = 0.97). The limit of detection (LoD) and limit of quantification (LoQ) were both determined as 0.01 ng *C. lupini* DNA/reaction (Table 1).

### 2.2. Quantification of C. lupini DNA from Seeds

*Colletotrichum lupini* DNA was detected in all samples of the three seed batches harvested from infested fields. Samples from the certified seed batch showed no amplification of *C. lupini* DNA. Seed batches W39, W63 and L9 contained 3.6 × 10^2^, 1.1 × 10^3^ and 1.4 × 10^4^ fg DNA/mg dry seed, respectively, with the DNA load of L9 being significantly higher than that of W39 (Figure 1). These results are in line with respective yield data and disease score indices observed for the mother plants of these seed batches in the field with high disease pressure. Seed batch L9, with the highest amount of *C. lupini* DNA, showed a high DSI (8) and the lowest yield (6.1 dt/ha), whereas seed batch W39, with the lowest amount of *C. lupini* DNA, had the lowest DSI (6.5) and the highest yield (12.3 dt/ha).

### 2.3. Quantification of C. lupini DNA from Stem, Leaves, and Roots

The DNA from three biological replicates was used to quantify *C. lupini* DNA in stem, leaf, and root samples of 29-day-old plants (Table 2). The mean relative lesion size was 18.2% and 0% and the mean disease score index was 3 and 1 for inoculated and control plants, respectively. Quantification of *C. lupini* DNA showed 1.7 × 10^4^ and 2.0 × 10^4^ fg DNA/mg dried shoot for the stem samples at the inoculation site and at the symptomless site 1 cm below the inoculation site, respectively. The symptomless stem sites 1 cm above the inoculation site also gave a reproducible positive signal; however, this was below the pre-defined LoQ level. Control samples and samples taken from root and leaf of inoculated plants did not result in amplification of *C. lupini* DNA.

## 3. Discussion

The qPCR assay developed in this study was shown to be *C. lupini* species-specific and successfully differentiated infected and certified seed batches with amplification efficiencies and LoD/LoQ levels similar to those reported previously [21,22]. The GAPDH gene was shown to be suitable to achieve species-specificity and has been widely used in *Colletotrichum* spp. diversity studies [7,9,23]. The method can now be used to assess the relationship between seed infection level and subsequent disease severity and yield in the field. A classical method to detect *C. lupini* on seeds is to morphologically identify appressoria after incubation on nutrient poor medium, which is applicable for smaller amounts of seeds as single seeds are assessed microscopically [18]. Additionally, an extended incubation period of 10–21 days is needed with potentially two assessments. Previously, it was demonstrated that pre-incubation of seeds for 72 h can enrich *C. lupini* biomass and, subsequently, the pathogen could be identified by PCR with a lower detection limit compared with the appressoria test [19]. The qPCR assay presented in this study can be used directly on seeds without incubation and allowed us to quantify the actual pathogen load per seed. Additionally, the qPCR assay is suitable as a high-throughput method for mass screening. Nevertheless, a combination of both enrichment of *C. lupini* biomass and qPCR could be explored to further increase sensitivity.

The disease cycle and dissemination of *C*. *lupini* within plant tissue is still under debate [23,24]. Our results suggest that the systemic spread of *C*. *lupini* in the vascular system was limited as, at 15 days post-inoculation, DNA of the pathogen was only detected at and below the inoculation side but was not detectable in leaves or roots. Thus, rapid infections of other plant parts might be caused by secondary processes such as rain splashes of nearby infected plants [12]. The qPCR method presented here can be used to further explore the disease cycle of *C*. *lupini* and will further add to our understanding of this devastating disease.

In summary, we demonstrated that our qPCR assay is a fast, practical, and sensitive method for the quantification of *C. lupini* DNA in white lupin seeds. The results reveal that the molecular quantification corresponds with the assumed pathogen inoculum level in seeds, making it a useful tool to assess lupin seed batches for seed health and their suitability for cultivation. This molecular assay can now be used to investigate how pathogen inoculum levels in seeds relate with disease outbreaks in the field in order to avoid the uncontrolled spread of this devastating seed-borne pathogen.

## 4. Materials and Methods

### 4.1. Fungal Strains

*C*. *lupini* and closely related strains known to colonize *Lupinus* spp. were used to validate the qPCR specificity (Table 1). *C. lupini* JA01 (of the globally distributed, highly genetic group II [8]) was obtained from the field in Mellikon, Switzerland. *C. trifolii* CTR3 was kindly provided by Roland Kölliker, ETH Zurich. All other fungal strains were retrieved from the Fungal Biodiversity Center (CBS-KNAW, Utrecht, The Netherlands).

### 4.2. Seed Material

White lupin seeds from the commercial variety Feodora were harvested from anthracnose-affected field experiments conducted in Full-Reuenthal, Switzerland in 2020 and, as a reference, certified seeds of the same variety harvested in France in 2015 were used. The harvested seed batches were categorized according to the anthracnose disease score index (DSI) assessed for the mother plants on the field at the time of flowering, as described by Alkemade et al. [25], with DSI values of 6.5, 8, and 8 for the infected seed batches W39, W63, and L9, respectively (Appendix A). Yield data for the field plots of the three seed batches W39, W63, and L9 were 12.3 dt/ha, 10.5 dt/ha, and 6.1 dt/ha, respectively.

### 4.3. DNA Extraction from Seeds and Fungal Cultures

Seed samples for DNA extractions consisted of 5 × 10 seeds, randomly selected from each seed batch, and ground in a ball mill (MM 200, Retsch, Haan, Germany) for 30 s at 30 Hz. All seed DNA extractions were performed with the Quick-DNA Plant/Seed DNA MiniPrep kit from ZYMO Research Corp (US) according to the manufacturer’s instructions. Briefly, 100 mg of ground dried seed material was added to the ZYMO lysis buffer together with 3 mm ceramic beads (Sigmund Lindner GmbH, Warmensteinach, Germany) in a 2 mL self-standing screw cap tube and disrupted using a high-speed tissue lyser (Qiagen, Hilden, Germany) for 2 min at 30 Hz. After further steps of purification with silica columns of the kit, the DNA was eluted with 100 µL of sterile, ddH_2_O and kept at −20 °C for further analyses.

Genomic DNA of fungal pure cultures was isolated as described by Minas et al. [26]. Mycelia was grown on potato dextrose agar (supplemented with 200 mg/L tetracycline hydrochloride; Carl Roth, Karlsruhe, Germany) at 22 °C for 7 days in the dark and then collected with a cell spreader after adding 2 mL ddH_2_O. After centrifugation, 1 mL hexadecyltrimethyl ammonium bromide (CTAB) lysis buffer (2% CTAB (Carl Roth, Germany), 1% polyvinylpyrrolidone (PVP) K25 (Carl Roth, Germany), 100 mM Tris–HCl (Fisher, Germany), 50 mM EDTA (Fisher, Germany), and 1.4 M NaCl, pH adjusted to 8.0 and 0.2% β-mercaptoethanol (Carl Roth, Germany) added shortly before use) was added, mixed with 0.5 mm zirconia/silica beads (BioSpec Products, Bartlesville, OK, USA), and lysed for 1 min at 30 Hz in a high-speed tissue lyser (TissueLyser II, Qiagen, Shanghai, China). Tubes were incubated at 65 °C for 1 h. After adding 600 μL of chloroform:isoamylalcohol (24:1; Roth, Germany), tubes were centrifuged at 20,000 g for 5 min. The aqueous supernatant was mixed with 400 μL ice-cold isopropanol. After 5 min of incubation at room temperature, tubes were centrifuged at 20,000 g for 5 min. After removal of the supernatant, pellets were washed twice with 200 μL of ice-cold 70% EtOH. Pellets were air dried and resuspended overnight in 100 μL double-distilled water (ddH_2_O) at 4 °C.

### 4.4. Primer Design and Amplification Protocol

A specific pair of primers and a fluorogenic hydrolysis probe (TaqMan), targeting the glyceraldehyde-3-phosphate dehydrogenase (GAPDH) gene of *C. lupini* (JQ948485.1; [7]), were designed with the Beacon Designer tool V8.16, (PREMIER Biosoft, Palo Alto, CA, USA) to amplify a 106-bp fragment (Appendix A; Aliview V1.26 [27]: Clup_GAPDH_F 5′-CCCACGGCAAAAGAGTCAGA-3′, Clup_GAPDH_R 5′-CGGCTGTTTCGGCATGATTG-3′, and the probe Clup_GAPDH_P 5′-FAM6-CGTCGTGTCATTACAACAAGCC -BHQ1-3′ were synthesized and HPLC-purified by Microsynth (Balgach, Switzerland). The specificity of the designed primer set was confirmed with BLASTN searches against the nucleotide collection on the NCBI webpage (https://blast.ncbi.nlm.nih.gov, accessed on 5 March 2020).

Each 20 µL reaction was comprised of 1 µL of DNA template (containing 10 ng DNA in the case of DNA from fungal pure cultures), 300 nM of primers Clup_GAPDH_F and Clup_GAPDH__R, 100 nM of probe Clup_GAPDH_P, and 10 µL of KAPA PROBE FAST qPCR Master Mix 2X (Kapa Biosystems Pty, Cape Town, South Africa). The amplification conditions were: 45 cycles with denaturation at 95 °C for 5 sec and annealing and elongation at 69 °C for 20 sec after an initial denaturation of 3 min at 95 °C using a Rotor Gene Q cycler (Qiagen, Hilden, Germany). The fluorescence was recorded after each annealing and elongation step with the green filter set (excitation: 470 nm, emission: 510 nm). As standards, DNA isolated from *C*. *lupini* was used at concentrations of 0.1, 1, and 10 ng/µL. The limit of detection (LoD) and the limit of quantification (LoQ) were determined by 1:10 serial dilutions of 10 ng/µL *C*. *lupini* DNA. The LoD represents the lowest concentration that is measurable and produces at least 95% positive replicates [28]. The LoQ is defined as the lowest concentration with a coefficient of variance <0.35 [20].

### 4.5. Quantification of C. lupini DNA in Lupin Plant Material

The plant growth conditions and sampling were as according to Alkemade et al. [25]. Briefly, pots were placed in a growth chamber (25 ± 2 °C, 16 h light, and ~70% relative humidity) for 14 days prior to inoculation. *C*. *lupini* JA01 was grown on potato dextrose agar for 6 to 8 days at 22 °C in the dark. Spores were harvested with a sterile spreader after flooding the Petri dish with 2 mL sterile ddH_2_O. The concentration was determined using a hemocytometer (0.1 mm; Marienfeld, Germany) and adjusted to 10^5^ spores/mL. Stem inoculation was performed by mildly puncturing the apical main stem with a sterile syringe needle followed by the application of 5 μL spore suspension [25]. Control plants were inoculated with sterile ddH_2_O. Both treatments included three biological replicates. After inoculation, plants were incubated for 48 h at 100% relative humidity (16 h light, 22 °C). After incubation, plants were placed back in the growth chamber. Fifteen days post-inoculation, disease symptoms were assessed by relative lesion size (percentage of overall stem length) and a disease score index of 1 to 9 (according to Alkemade et al. [25]) before about 150 mg of stem, leaf, and root were collected. Stem samples were taken at the stem inoculation site and at symptomless stem sites 1 cm above and below it. DNA of *C. lupini* was extracted and quantified as described above.

### 4.6. Statistical Analyses

Statistical analyses were performed with the program JMP V14.1.0 (SAS Institute, Cary, NC, USA). DNA concentrations were log_10_-transformed to satisfy the assumption of normality of residuals and homogeneity of variance. In order to test for significant differences in the DNA concentration of *C. lupini* between lupin seed batches the Tukey-HSD (honestly significant difference) test (*p* ≤ 0.05) was applied.

## Figures and Tables

**Figure 1 plants-10-01548-f001:**
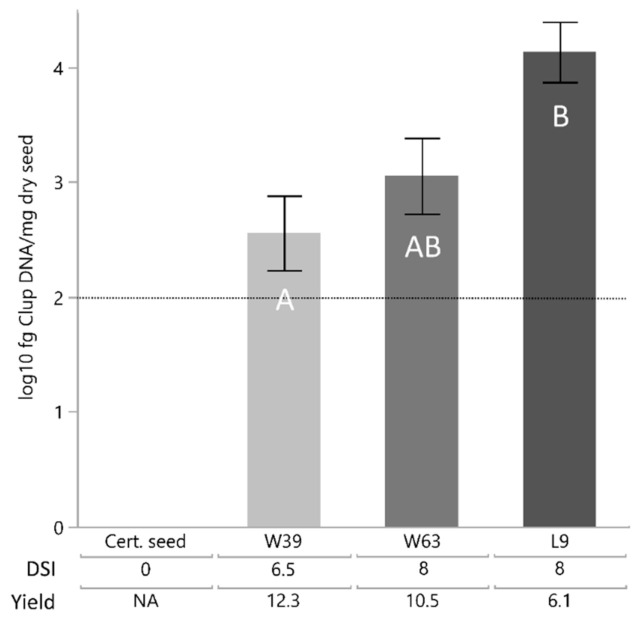
Mean DNA concentration of *C*. *lupini* (Clup) in four seed batches (including a certified seed batch) with different disease score indices (DSI) and yields (in dt/ha) of the mother plants. The dashed horizontal line indicates the limit of quantification. Error bars indicate the standard error of the mean. Mean values with a common letter are not significantly different (Tukey-HSD, *p* < 0.05).

**Table 1 plants-10-01548-t001:** Specificity, sensitivity, and efficiency of the *C*. *lupini* qPCR assay. Quantification cycle (Cq) values (+/− standard deviation) are given for four 10-fold dilutions of *C. lupini* DNA standards for six different *Colletotrichum* spp., four of which belong to different clades of the *C. acutatum* species complex. R^2^ is the coefficient of correlation between Cq and DNA standard values. Sensitivity is expressed via the limit of detection and quantification (equal in this case).

Clade	Species	Isolate		Cq Value			R^2^ Value	Efficiency	LoD/LoQ
			10 ng/µL	1 ng/µL	0.1 ng/µL	0.01 ng/µL			
1	*C*. *lupini*	JA01	24.6 +/−0.41	28.5 +/−0.51	31.9 +/−0.42	35.5 +/−0.72	0.97	1.06	0.01 ng/µL
1	*C*. *tamarilloi*	CBS 129814	ND	ND	ND	ND			
2	*C*. *nymphaeae*	CBS 130239	ND	ND	ND	ND			
4	*C*. *acutatum*	CBS 369.73	ND	ND	ND	ND			
-	*C*. *coccodes*	CBS 641.97	ND	ND	ND	ND			
-	*C*. *trifolii*	CTR3	ND	ND	ND	ND			

ND = below detection limit.

**Table 2 plants-10-01548-t002:** Quantification of *C. lupini* DNA in different plant organs of artificially inoculated plants 15 days post-inoculation.

Plant Organ	Inoculated Plants	Control Plants
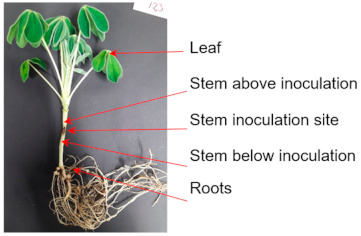	*C. lupini* DNA/mg dry plant material	
	ND	ND
	ND	ND
	1.7 × 10^4^ fg	ND
	2.0 × 10^4^ fg	ND
	ND	ND

ND = below detection limit.

## Data Availability

All data, tables and figures in this manuscript are original. Raw data of this study is available upon request.

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
