# Peer review of "A qPCR Assay for the Fast Detection and Quantification of Colletotrichum lupini"

_plants, 2021, doi:10.3390/plants10081548_

Round 1
Reviewer 1 Report
The authors developped a qPCR tool to detect Colletotrichum lupini in dry seeds and in shoots.
major comments :
1/the verification of the specificity of the tool is incomplete since 2 other closely related species were not tested (C. paranaense and C. abscissum) in qPCR, moreover the alignments of the primer/probe set on the GAPDH gene of these species was not shown, which raises questions about the specificity of the tool.
2/on the other hand the detection on seedling is not conclusive: in these conditions on this lupin genotype, lesions of several mm should have been observed since the Feodora variety is considered as sensitive, the detection under the inoculation point with the same load of fungus between the inoculation point and 1cm under the inoculation point seems rather to show that the biotest has failed. A detection further away from the inoculation point would have been more convincing. Perharps you should use a stronger inoculum such as in Dubrulle 2020 (10e7 spores/ml) ? On the other hand the detection of the fungus at the inoculation point does not show that the fungus is still alive, only a PCR using a live/dead method with e.g. propidium monoazide is more suitable.
minor comments:
line 161 : ref 28 and not 11
line 198 : database not precised , did you use nucleotide collection or wgs ?
line 213 : the number of plants for each biological replicate is not specified
line 263 : reference 8: may be inappropriate as not yet published
Author Response
Major comments:
1. the verification of the specificity of the tool is incomplete since 2 other closely related species were not tested (C. paranaense and C. abscissum) in qPCR, moreover the alignments of the primer/probe set on the GAPDH gene of these species was not shown, which raises questions about the specificity of the tool.
Authors’ repsonse:
These species were not included to the specificity validation of qPCR because we did not have them available in our lab and because they, among others included in the alignment, although closely related to C. lupini have never been never reported as lupin pathogens or detected from lupin plant material and we therefore did not consider them to compromise the specificity of a qPCR test that aims to specifically quantify C. lupini in lupin. Nevertheless, we did include them for completeness to the alignment showing , see modified Figure S1 and Chapter 4.1. The two species in question show a mismatch at the critical 3’ end of the forward primer.
2. on the other hand the detection on seedling is not conclusive: in these conditions on this lupin genotype, lesions of several mm should have been observed since the Feodora variety is considered as sensitive, the detection under the inoculation point with the same load of fungus between the inoculation point and 1cm under the inoculation point seems rather to show that the biotest has failed. A detection further away from the inoculation point would have been more convincing. Perharps you should use a stronger inoculum such as in Dubrulle 2020 (10e7 spores/ml)? On the other hand the detection of the fungus at the inoculation point does not show that the fungus is still alive, only a PCR using a live/dead method with e.g. propidium monoazide is more suitable.
Authors’ response:
Thank you for the scrutiny. By mistake, the picture in Table 2 showed an uninoculated control plant. The picture was replace by an inoculated plant showing a 0.9 cm lesion. We also added information on lesion size and overall disease score to the manuscript in Chapters 2.3 and 4.5. The authors agree that the method can be expanded to also consider live/dead information, however, this is beyond the scope of our study. Our results show that the pathogen can be detected in symptomless stem material incidacting the presence of actively growing hyphae, which makes it a suitable tool for initial disease cycle assessments.
Minor comments:
line 161 : ref 28 and not 11
- This was corrected, line 167
line 198 : database not precised , did you use nucleotide collection or wgs ?
- Information about the database was added, line 203
line 213 : the number of plants for each biological replicate is not specified
-The number of plants was added, line 227
line 263 : reference 8: may be inappropriate as not yet published
-In the meantime the paper was published, line 275-276 (not in track-change mode due to Endnote text function)
Reviewer 2 Report
Authors developed qPCR method for detection of important pathogen of white lupi - Colletotrichum lupini.
The paper and study as whole sound well thought and designed, however I have some suggestions and issuses:
- I would like to see amplification plots, even as a supplementary materials, of at least reactions to determine limit of detection - lowest quantity detected was in a weak positive range of Ct, so it would be very impactful to see how the reaction behaved.
- Did you consider testing lower concentration of input DNA? Your study design seem to present some headroom before reaching Ct ~37. My prediction is that you could achieve at least two times ( about 5 pg/ul) lower LoD.
- You inconsistently use italics in names of genera, species and complexes. This have to be corrected.
- You should double check the manufacturers data of reagents and equipment you use - for example you say that KAPA kit is produced by Sigma, which is not true - Sigma distributes the kit, but KAPA is currently owned by Roche. Moreover you inconsistently use ISO 3166-1 alpha-2 two letter codes and full names of country for identifying the manufacturer.
Author Response
Authors developed qPCR method for detection of important pathogen of white lupi - Colletotrichum lupini.
The paper and study as whole sound well thought and designed, however I have some suggestions and issues:
1. I would like to see amplification plots, even as supplementary materials, of at least reactions to determine limit of detection - lowest quantity detected was in a weak positive range of Ct, so it would be very impactful to see how the reaction behaved.
Authors’ response:
As described in Chapter 4.4, the limit of detection was defined according to Forootan et al. (2017) based on reproducible (>95%) positive signals. A positive signal was found when the signal exceeds the background level (threshold level computed by the manufacturer’s (Qiagen) software). This is basic, common knowledge of the qPCR methodology. We therefore believe that an additional figure does not provide information sufficiently relevant for publication.
2. Did you consider testing lower concentration of input DNA? Your study design seems to present some headroom before reaching Ct ~37. My prediction is that you could achieve at least two times (about 5 pg/ul) lower LoD.
Authors’ response:
We used a 10x serial dilution of DNA (from 10 to 0.00001 ng/μl ) for sensitivity testing and show in Table 1 only the lowest concentration of 0.01 ng/μl, which still led to a reprocible amplification. We agree with the reviewer that an LOD between 0.01 and 0.001 pg/ul might be possible. For consistency and also because we currently do not have resources to produce and assess new standards, we kept 0.01 ng/μl as LOD as this was the last 10x dilution step with an amplification.
3. You inconsistently use italics in names of genera, species and complexes. This have to be corrected.
Authors’ response:
This was corrected throughout the document.
4. You should double check the manufacturers data of reagents and equipment you use - for example you say that KAPA kit is produced by Sigma, which is not true - Sigma distributes the kit, but KAPA is currently owned by Roche. Moreover, you inconsistently use ISO 3166-1 alpha-2 two letter codes and full names of country for identifying the manufacturer.
Authors’ response:
We believe it is more useful to the reader to indicate where the product can be purchased rather than where it was produced. We would consider the latter if Roche also distributed the kit, which, to the best of our knowledge, is not the case.
The country codes were corrected throughout Chapter 4.
Reviewer 3 Report
Dear authors,
The manuscript is well written and has a significant impact on Colletotrichum detection and management in field conditions.
The developed assay is fine. However, I want to see one extra simple experiment:
Please use a standard PCR and repeat the PCR 3 times. Use a 20ul PCR reaction for each of the diluted genomic DNA samples and for the seed batch isolated DNA with a regular enzyme and at least 35, better 40 cycles. Run the 20ul on a gel, your 106bp fragment should be nicely visible. What are the detection limits compared to the Taqman assay? My point being, if a regular PCR would be as sensitive as the qPCR in detecting the infection it would not be necessary to use an expensive Taqman assay.
Author Response
Thank you for your valuable suggestion. The forward and reverse primers with the hydrolysis probe did not lead to species specificity. We therefore cannot perform a standard PCR assay.
Reviewer 4 Report
This paper addresses a practical problem with a rigorous scientific approach. It is essential but contains all needed information. The style is clear. I added just few minor comments and text editing (see notes in the text, attached file)

Author Response
Minor modifications as suggested by the reviewer have been done throughout the document.